# Design, Synthesis and Bioactivity Evaluation of Coumarin–BMT Hybrids as New Acetylcholinesterase Inhibitors

**DOI:** 10.3390/molecules27072142

**Published:** 2022-03-26

**Authors:** Fanxin Zeng, Tao Lu, Jie Wang, Xuliang Nie, Wanming Xiong, Zhongping Yin, Dayong Peng

**Affiliations:** 1East China Woody Fragrance and Flavor Engineering Research Center of National Forestry and Grassland Administration, College of Forestry, Jiangxi Agricultural University, Nanchang 330045, China; zengfanxin0629@163.com (F.Z.); taozi79883251@163.com (T.L.); 2Jiangxi Academy of Forestry, Nanchang 330032, China; 3Key Laboratory of Chemical Utilization of Plant Resources of Nanchang, College of Chemistry and Materials, Jiangxi Agricultural University, Nanchang 330045, China; wangjie951127@163.com (J.W.); 13576953044@163.com (X.N.); xiongwm10@163.com (W.X.); 4Jiangxi Key Laboratory of Natural Products and Functional Food, Jiangxi Agricultural University, Nanchang 330045, China

**Keywords:** coumarin–BMT hybrids, acetylcholinesterase inhibitors, bioactivity evaluation, molecular docking

## Abstract

Coumarin possesses the aromatic group and showed plentiful activities, such as antioxidant, preventing asthma and antisepsis. In addition, coumarin derivatives usually possess good solubility, low cytotoxicity and excellent cell permeability. In our study, we synthesized the compound bridge methylene tacrine (BMT), which has the classical pharmacophore structure of Tacrine (THA). Based on the principle of active substructure splicing, BMT was used as a lead compound and synthesized coumarin–BMT hybrids by introducing coumarin to BMT. In this work, 21 novel hybrids of BMT and coumarin were synthesized and evaluated for their inhibitory activity on AChE. All obtained compounds present preferable inhibition. Compound **8b** was the most active compound, with the value of K*_i_* as 49.2 nM, which was higher than Galantamine (GAL) and lower than THA. The result of molecular docking showed that the highest binding free energy was −40.43 kcal/mol for compound **8b**, which was an identical trend with the calculated K*_i_*.

## 1. Introduction

Alzheimer’s disease (AD) was described by a German neuropathologist, Alois Alzheimer, in 1906, which is a neurodegenerative disorder associated with a progressive and rather irreversible decline in memory and various other cognitive capabilities [1,2]. It has become one of the “epidemics” in the world, the fourth most common cause of death in developed countries. According to expert predictions of AD [3], the number of dementia patients globally will grow from 36 million in 2010 to 66 million in 2030. By 2050, the population is being speculated to rise to 115 million. This means that in the near 30–40 years, almost each person would be affected by dementia throughout their life cycle either as patients or as their caregivers [4]. At present, the causal mechanisms of progressing AD are still not well defined. However, scientists have proposed several theories, such as loss of cholinergic function (known as cholinergic hypothesis), the amyloid cascade (amyloid hypothesis), oxidative stress, decrease in steroid hormone concentration and inflammation process [5,6].

Acetylcholine (ACh) is a key neurotransmitter in the cholinergic hypothesis. In AD brains, it is decreased both in concentration and function. Acetylcholinesterase (AChE) is responsible for hydrolyzing ACh, which can regulate the level of ACh in the AD brains. AChE inhibitors are the medicine used for the clinical treatment of AD. However, these agents do not cure the progression of disease but only mask symptoms. There are five AChE inhibitors which have been employed in treatment of AD: Rivastigmine, (-)-HuprineA, Galantamine (GAL), Donepezil and Tacrine (THA) [7,8]. These agents can effectively relieve the symptoms of AD, whereas there are various side effects in clinical treatment, for example, flatulence, nausea, emesis and transaminase elevation [9,10]. Therefore, there is an urgent need for developing potent AChE inhibitors to treat AD without severe side effects.

THA is the most effective acetylcholinesterase inhibitor (Figure 1). However, THA exhibits a well-established hepatotoxicity with liver cell necrosis and an oxidative stress which reduce its clinical use [11]. Harel et al. [12] briefly explored the inhibition mechanism of THA and AChE by the molecular docking method. Their results showed that the pyridine ring of THA forms a “sandwich” type of π–π stacking with the aromatic residues Trp84 and Phe330 of the CAS site in the AChE structure (*Torpedo* acetylcholinesterase, EC 3.1.1.7). In addition, it forms a hydrogen bond of 3.2 Å with His440 (one of the catalytic triads). These conclusions laid the foundation for the design of novel AChE inhibitors with both potently inhibitory action and little side effects. To discover new THA derivatives with higher activity and multiple functions, a variety of hybrids have been synthesized, such as THA–THA hybrids (Figure 1a) [13], donepezil–tacrine hybrids (Figure 1b) [14], benzoates (or phenylacetates or cinnamates)–tacrine hybrids (Figure 1c) [15], and tacrine–8-hydroxyquinoline hybrids (Figure 1d) [16,17]. Many of them show better antioxidant properties, more potent inhibitory properties and lower cell toxicity than THA [18]. Studies have shown that the development of hybrids of THA spliced with other compounds, such as new acetylcholinesterase inhibitors, is of great research significance [8,19].

Coumarin possesses the aromatic group and shows plentiful activities, such as antioxidant potential and preventing asthma and antisepsis [20,21,22]. In addition, coumarin derivatives usually possess good solubility, low cytotoxicity and excellent cell permeability. In our study, we synthesized the compound bridge methylene tacrine (BMT), which has the classical pharmacophore structure of THA. Based on the principle of active substructure splicing, BMT was used as a lead compound and synthesized coumarin–BMT hybrids by introducing coumarin to BMT. The inhibitory ability of coumarin–BMT hybrids on AChE was determined. The further structure–activity relationship analysis provided a reference for the development of functional food factors and drugs for treatment of AChE.

## 2. Results and Discussion

### 2.1. Chemistry

In this study, twenty-one hybrids of coumarin–BMT were synthesized. All of them were reported for the first time. The general route to the synthesis of the hybrids was illustrated in Figure 2. First, 4,11,11-trimethyl-1,2,3,4-tetrahydro-1,4-methanoacridin-9-amine (**1a**) was synthesized by following known procedures [23,24]. Compound **1a** was treated with NaNO_2_, SnCl_2_, and diluted with hydrochloric acid at r.t. to obtain 9-chloro-4,11,11-trimethyl-1,2,3,4-tetrahydro-1,4-methanoacridine (**5a**) (Yield:30%), and by-product 4,11,11-trimethyl-1,2,3,4-tetrahydro-1,4-methanoacridin-9-ol was obtained; yield is 55%. Surprisingly, we did not separate the product, dried it with magnesium sulfate and chlorinated it with phosphorus oxychloride, and obtained **5a** in 81% yield. At the same time, the crystal structure of **5a** was obtained. The commercial phenol derivatives were treated with Et_3_N, MgCl_2_ and paraformaldehyde in acetonitrile at reflux to obtain intermediates **2a**~**2g**. These intermediates were then treated with diethyl malonate, piperidine and acetic acid in EtOH at reflux to obtain intermediates **3a**~**3g** [25]. These intermediates were then treated with 10% NaOH at reflux to obtain coumarin derivatives. These coumarin derivatives were then treated with BMT derivatives, which were gained from the halogenation and diamines substitution reaction of BMT, to obtain coumarin–BMT hybrids.

### 2.2. Biological Evaluations

#### 2.2.1. Crude Screening on AChE Inhibition

To determine the potential of the target compounds **7a**~**9g** for the treatment of AD, their AChE inhibitory activity was determined by the method of Elman et al. [26]. The inhibition activities against AChE at 2 μM were summarized, as shown in Table 1. The results indicated that all the tested compounds **7a**~**9g** showed significant AChE inhibitory activity at this dosage. Compound **8e**, with a -OMe group, showed the most potent inhibition for AChE, with an inhibition ratio of 87.48%, which was higher than that of the positive control Galantamine (GLA) (69.23%) and close to THA (95.75%).

The optimal chain length between coumarin and BMT units determined experimentally for inhibiting AChE were five (**7a**~**7g**) and six (**8a**~**8g**) methylene groups. The inhibitory effect of coumarin–BMT hybrids with seven methylene groups (**9a**~**9g**) was decreased prominently. The 5- and 6-methylenes linker seemed to the best suitable length for AChE inhibition. This conclusion is consistent with our previous paper.

With the same chain length, the inhibition activity of hybrids **7e**, **8e**, **9e** (R^1^ is a -OMe group) for AChE was slightly better than that for hybrids **7b**, **8b**, **9b** (R^1^ is a -Me group), and was significantly better than other hybrids. It indicated that hybrids with different substituent groups display different inhibitory activity and hybrids with methoxy group or methyl group display better inhibitory activity than others, as shown in Figure 3. It speculated that the introduction of electron donating groups to coumarin–BMT hybrids, such as methoxy group or methyl group, at position 6 or 7 can improve inhibitory activity. On the other hand, the introduction of electron withdrawing groups, such as -Br or -OCF_3_ group, can lower inhibitory activity.

These characters provide theoretical foundation for developing a better AChE inhibitor of coumarin–BMT hybrids by further optimization of compounds.

#### 2.2.2. Kinetic Characterization of AChE Inhibition

The inhibition of AChE by several representative compounds **7b**, **7e**, **7f**, **7g**, **8b**, **8e** was further investigated using graphical analysis of steady state inhibition data, as shown in Figure 4. The Lineweaver–Burk plots showed both increasing slopes and increasing intercepts for higher inhibitor concentration. As listed in Table 2, compounds **8b** (K*_i_* = 49.20 nM) and **8e** (K*_i_* = 50.81 nM) were the most potent AChE inhibitors among all coumarin–BMT hybrids tested. However, their K*_i_* values were both slightly higher than the K*_i_* value of THA (K*_i_* = 31.13 nM), and lower than the K*_i_* value of GAL (K*_i_* = 61.93 nM). Their pattern indicated a mixed-type inhibition, which was similar to that of THA and GAL.

Regarding the influence of the linker, Coumarin–BMT hybrids **8b** and **8e** with six methylene groups were more potent AChE inhibitors than those with five (**7b,** K*_i_* = 189.82 nM; **7e**, K*_i_* = 65.13 nM; **7f**, K*_i_* = 118.34 nM; **7g**, K*_i_* = 152.61 nM) methylene groups. It indicated that the suitable linker length is conductive for compounds to enter the active site of protein. Regardless of the influence of the linker, Coumarin–BMT hybrids with the methoxy group on position 6 (**7e**) were more potent AChE inhibitors than that on position 7 (**7f**). The result show that the steric hindrance on position 7 may be unfavorable on the activity of compounds.

#### 2.2.3. Molecular Docking, MD Simulation and Binding Energy Calculation

To understand the structure–activity relationship at the atomic level, we performed the molecular docking and binding free energy calculations for the compounds **7b** and **8b**. The corresponding root mean square deviation (RMSD) was an evaluative criterion to estimate the equilibration of the backbone atoms along the MD trajectories. The smaller the amplitude of the fluctuation is, the more stable the system is. The RMSD values of compounds **7b** and **8b** were computed, respectively. As we can see from Figure 5, the RMSD values of **7b** tend to be convergent after 10 ns of simulation, while the RMSD values of **8b** tend to be convergent after 30 ns of simulation, showing that the whole systems, in general, were equilibrated. Finally, 50 ns production MD simulation was performed.

The details about the energy calculation are show in Table 3. The highest binding free energy (ΔGbind) was −37.48 kcal/mol for compound **7b**, and the lowest binding energy was −40.43 kcal/mol for compound **8b**. The corresponding experimental binding affinity (K*_i_*) was 49.20 nM for compound **8b** and 189.82 nM for compound **7b**. Moreover, the experimental binding affinity showed an identical trend with the calculated binding energy. It confirmed the reliability of the computational models constructed in this work. Computational simulations revealed that the inhibitor **7b** interacted through hydrogen bonds with Tyr334, Asp446 and Glu449, and that the inhibitor **8b** interacted through hydrogen bonds with Trp436 and π–π interactions with Trp235, Phe335, Trp81 and Phe294 (Figure 6). This indicate that these sites are the key residues for inhibitors binding.

The components of binding energy were calculated by MM-GBSA method. It could be seen from Table 3 that the hydrogen bond was the major favorable contribution on the binding energies between compound **8b** and AchE. The results suggest that the carbon chain was in favor of the stability of the Coumarin–BMT hybrids-AchE system. Perhaps the optimal chain length is six. This provides a useful guide for further research on Coumarin–BMT hybrids.

## 3. Conclusions

In summary, 21 Coumarin–BMT hybrids were synthesized targeting for the inhibition of AChE. Most of the compounds synthesized exhibited potent bioactivities, with an inhibition ratio over 50% on a concentration of 2 μM, in which the most potent compounds **8b** and **8e** have an inhibition ratio of 86.60% and 87.48%, respectively. The electron donating substituted hybrids showed higher inhibitory effects on AChE, and compounds **8b** and **8e** with six methylene groups had the best AChE inhibitory. The inhibition kinetics of **8b** and **8e** were analyzed using Lineweaver–Burk plots, which revealed that the compounds were a mixed-type inhibitor and could bind to both the CAS and PAS of AChE. These results indicate that the Coumarin–BMT scaffold is a novel chemotype for developing new AChE inhibitors, which offers potential for the discovery of anti-AD drugs.

## 4. Experimental Section

### 4.1. Chemistry

^1^H and ^13^C NMR spectra were recorded using TMS as the internal standard in CDCl_3_ and DMSO with a Burk BioSpin GmbH spectrometer at 500 MHz and 125 MHz, respectively. High resolution mass spectra (HRMS) were recorded on Shimadzu. Silica gel (200–300 mesh) purchased from Yantai Xincheng Chemical Co. Ltd. Melting points (mp) were determined using an SRS-OptiMelt automated point instrument without correction.

### 4.2. The Synthesis of BMT

To a mixture of camphor (10 mmol) and 2-aminobenzonitrie (10 mmol) in toluene (200 mL), anhydrous aluminum chloride (13.3 g) was added. The reaction system was stirred at 140 °C and refluxed overnight via water separator, then cooled at room temperature. The precipitate was isolated by filtration and the yellow solution was consecutively added 2 M NaOH (80 mL). The mixture solution was stirred at 110 °C, refluxed overnight. The resulting mixture was cooled at room temperature, then extracted with CH_2_Cl_2_ three times. The organic layer was washed with saturated NaCl solution, dried over anhydrous MgSO_4_, concentrated in vacuo, re-crystallized in toluene, then filtered to afford BMT as a solid.

### 4.3. The Synthesis of Coumarin Derivatives

#### 4.3.1. General Procedures for the Preparation of Intermediate **2a**~**2g**

A mixture of phenolic derivatives (24.2 mmol), Et_3_N (92.1 mol), anhydrous MgCl_2_ (37 mmol) and Paraformaldehyde (167 mmol) in Acetonitrile (100 mL) was stirred at 95 °C and refluxed overnight. The mixture solution was cooled at room temperature and was adjusted to be acidic (pH = 2) with HCl solution. The resulting mixture was filtrated and the solution was extracted with CH_2_Cl_2_ three times. The organic layer was dried over anhydrous MgSO_4_, concentrated in vacuo and purified by flash column chromatography with petroleum ether (PE).

#### 4.3.2. General Procedures for the Preparation of Intermediate **3a**~**3g**

To a mixture of intermediate **2b**~**2g** (10 mmol), diethyl malonate (12 mmol) in EtOH (120 mL), piperidine (1 mL) and acetic acid (0.2 mL) were added. The reaction system was stirred at 95 °C overnight. The mixture solution was cooled at room temperature, then H_2_O (100 mL) was added. Recrystallisation from the mixture solution (still standing at 0 °C) yielded a crud solid of intermediate **3a**~**3g**. The crud solid was washed with 50% cooled EtOH and concentrated in vacuo to obtain a white solid.

#### 4.3.3. General Procedures for the Preparation of Coumarin Derivatives **4a**~**4g**

To a solution of intermediate **3a**~**3g** in EtOH (20 mL), 10% NaOH (20 mL) was added. The reaction system was stirred at 95 °C for 2 h, then cooled at room temperature and adjusted to be acidic (pH = 2) with HCl solution. Recrystallisation from the mixture solution (still standing at 0 °C) obtained crud solid of coumarin derivatives. The crud solid was washed with cooled H_2_O, and concentrated in vacuo to obtain white solid.

2-oxo-2H-chromene-3-carboxylic acid (**4a**): White solid; yield 54%. ^1^H NMR (500 MHz, DMSO) δ 13.30 (s, 1H), 8.75 (s, 1H), 7.91 (dd, *J* = 7.7, 1.5 Hz, 1H), 7.74 (m, 1H), 7.42 (s, 2H).

6-methyl-2-oxo-2H-chromene-3-carboxylic acid (**4b**): Yellow solid; yield 45%. ^1^H NMR (500 MHz, DMSO) δ 13.12 (s, 1H), 8.67 (s, 1H), 7.70 (d, *J* = 1.3 Hz, 1H), 7.59–7.53 (m, 1H), 7.35 (d, *J* = 8.5 Hz, 1H), 3.34 (s, 3H).

6-bromo-2-oxo-2H-chromene-3-carboxylic acid (**4c**): White solid; yield 48%. ^1^H NMR (500 MHz, DMSO) δ 13.44 (s, 1H), 8.73 (s, 1H), 8.21 (d, *J* = 2.4 Hz, 1H), 7.91 (dd, *J* = 8.8, 2.4 Hz, 1H), 7.46 (d, *J* = 8.8 Hz, 1H).

2-oxo-6-(trifluoromethoxy)-2H-chromene-3-carboxylic acid (**4d**): White solid; yield 40%. ^1^H NMR (500 MHz, DMSO) δ 13.46 (s, 1H), 8.73 (s, 1H), 8.02 (d, *J* = 2.2 Hz, 1H), 7.73 (dd, *J* = 9.0, 2.3 Hz, 1H), 7.57 (d, *J* = 9.1 Hz, 1H).

2-oxo-6-(trifluoromethoxy)-2H-chromene-3-carboxylic acid (**4e**): Faint yellow solid; yield 42%. ^1^H NMR (500 MHz, DMSO) δ 13.00 (s, 1H), 8.74 (s, 1H), 7.84 (d, *J* = 8.7 Hz, 1H), 7.08–6.97 (m, 2H), 3.90 (s, 3H).

7-methoxy-2-oxo-2H-chromene-3-carboxylic acid (**4f**): Yellow solid; yield 43%. ^1^H NMR (500 MHz, DMSO) δ 13.24 (s, 1H), 8.72 (d, *J* = 16.6 Hz, 1H), 7.49 (d, *J* = 2.9 Hz, 1H), 7.40 (d, *J* = 9.1 Hz, 1H), 7.34 (dd, *J* = 9.1, 2.9 Hz, 1H), 3.83 (s, 3H).

6,7-dimethoxy-2-oxo-2H-chromene-3-carboxylic acid (**4g**): Yellow solid; yield 42%. ^1^H NMR (500 MHz, DMSO) δ 12.89 (s, 1H), 8.69 (s, 1H), 7.45 (s, 1H), 7.12 (d, *J* = 3.4 Hz, 1H), 3.91 (s, 3H), 3.81 (s, 3H).

### 4.4. The Synthesis of Coumarin–BMT Hybrids

#### 4.4.1. The Synthesis of Intermediate BMT Derivatives (**5a**)

A solution of BMT in HCl solution was stirred at 0 °C until BMT dissolved completely. Subsequently, H_2_O (12 mL) and 2.5M Sodium nitrite solution (6 mL) was added, stirred at 0 °C for 30 min, then the ice bath was removed and 4.1 M stannous chloride dihydrate solution (6 mL) was added slowly. The mixture solution was stirred at room temperature until the reaction was fully carried out. The resulting solution was neutralized by saturated NaHCO_3_ solution and extracted by CH_2_Cl_2_ three times. The organic layer was dried over anhydrous MgSO_4_, concentrated in vacuo and purified by flash column chromatography with PE/ ethyl acetate (EA). Faint yellow solid; yield 84%. ^1^H NMR (500 MHz, CDCl_3_) δ 8.15 (m, 1H), 8.08 (m, 1H), 7.69–7.62 (m, 1H), 7.55 (m, 1H), 3.25 (d, *J* = 4.1 Hz, 1H), 2.27–2.15 (m, 1H), 2.04–1.92 (m, 1H), 1.42 (s, 3H), 1.33 (ddd, *J* = 14.9, 12.3, 3.6 Hz, 2H), 1.08 (s, 3H), 0.61 (s, 3H). ^13^C NMR (125 MHz, CDCl_3_) δ 172.09, 147.60, 129.39, 129.00, 128.19, 127.58, 123.81, 119.90, 119.34, 55.28, 54.43, 47.47, 32.53, 25.63, 20.10, 19.15, 10.65. Complex (**5a**) is the orthorhombic system, space group C2221 with a= 1.25080(7) nm, b = 1.52427(9) nm, c = 1.52870(9) nm, α = 90°, β = 90°, γ= 90°, V = 2.9146(3) nm^3^ (Figure 7).

#### 4.4.2. General Procedures for the Preparation of Intermediate **6a**~**6c**

For a mixture of compound **5a** (3.68 mol), the appropriate diamines (1,5-diaminopentane, ethylenediamine and 1,7-diaminoheptane, 7.36 mmol), phenol (22.5 mmol) and NaI (0.75 mmol) were stirred overnight at 180 °C on the protection of N_2_. The resulting solution was washed by 2M NaOH solution, distilled water and saturated NaCl solution in turn, then extracted by CH_2_Cl_2_ three times. The organic layer was dried over anhydrous MgSO4, concentrated in vacuo and purified by flash column chromatography with PE/EA.

#### 4.4.3. General Procedures for the Preparation of Coumarin–BMT Hybrids **7a**~**9g**

To a solution of intermediate **6a**~**6c** (1 mmol) and coumarin derivatives **4a**~**4g** (1 mmol) in CH_2_Cl_2_ (20 mL), benzotriazole-1-yl-oxytripyrrolidinophosphonium hexafluorophosphate (PyBOP, 1.3 mmol) and triethylamine (2.6 mmol) were added. The reaction system was stirred at room temperature until the reaction was fully carried out, then washed by 10% citric acid solution, 10% NaHCO_3_ solution and distilled water in turn, extracted by CH_2_Cl_2_ three times. The organic layer was dried over anhydrous MgSO4, concentrated in vacuo and purified by flash column chromatography with PE/EA/triethylamine.

2-oxo-N-(5-((4,11,11-trimethyl-1,2,3,4-tetrahydro-1,4-methanoacridin-9-yl)amino)pentyl)-2H-chromene-3-carboxamide(**7a**): Intermediate **6a** was treated with intermediate **4a** according to general procedure to give the desired product **7a** as a white solid (42%). mp: 100–102 °C. ^1^H NMR (500 MHz, CDCl_3_) δ 8.91 (s, 1H), 8.89 (s, 1H), 7.96 (d, *J* = 8.3 Hz, 1H), 7.74 (d, *J* = 7.6 Hz, 1H), 7.71–7.63 (m, 2H), 7.53–7.48 (m, 1H), 7.43–7.31 (m, 3H), 4.52 (bs, 1H), 3.65–3.56 (m, 1H), 3.55–3.47 (m, 3H), 3.27 (d, *J* = 4.0 Hz, 1H), 2.20–2.12 (m, 1H), 1.92–1.84 (m, 1H), 1.82–1.69 (m, 4H), 1.63–1.55 (m, 2H), 1.37 (s, 3H), 1.33 (d, *J* = 8.3 Hz, 2H), 1.02 (s, 3H), 0.65 (s, 3H). ^13^C NMR (125 MHz, CDCl_3_) δ 173.23, 161.59, 161.52, 154.36, 148.34, 142.26, 134.04, 129.77, 129.39, 127.39, 125.30, 123.77, 120.00, 119.50, 118.61, 118.39, 118.01, 116.59,54.32, 53.55, 50.96, 45.98, 39.46, 32.08, 30.36, 29.36, 27.01, 24.34, 20.33, 19.18, 14.17, 10.86. HRMS(ESI) *m*/*z* [M + H]^+^: calcd for C_32_H_35_N_3_O_3_: 510.2712, found 510.2739.

6-methyl-2-oxo-N-(5-((4,11,11-trimethyl-1,2,3,4-tetrahydro-1,4-methanoacridin-9-yl)amino)pentyl)-2H-chromene-3-carboxamide(**7b**): Intermediate **6a** was treated with intermediate **4b** according to general procedure to obtain the desired product **7b** as a yellow solid (33%). mp: 131–132 °C. ^1^H NMR (500 MHz, CDCl_3_) δ 8.92 (s, 1H), 8.86 (s, 1H), 7.97 (d, *J* = 8.3 Hz, 1H), 7.74 (d, *J* = 8.3 Hz, 1H), 7.54–7.42 (m, 3H), 7.35 (t, *J* = 7.6 Hz, 1H), 7.30 (d, *J* = 8.3 Hz, 1H), 4.55 (bs, 1H), 3.66–3.56 (m, 1H), 3.55–3.48 (m, 3H), 3.27 (d, *J* = 4.0 Hz, 1H), 2.44 (s, 3H), 2.19–2.13 (m, 1H), 1.91–1.84 (m, 1H), 1.83–1.68 (m, 4H), 1.63–1.54 (m, 2H), 1.38 (s, 3H), 1.33 (d, *J* = 8.6 Hz, 2H), 1.02 (s, 3H), 0.64 (s, 3H). ^13^C NMR (125 MHz, CDCl_3_) δ 173.01,161.77, 161.74, 152.57, 148.34, 135.24, 135.20, 129.35,128.26, 127.47, 123.82, 119.96, 119.55, 118.38, 118.19, 117.94, 116.29, 54.36, 53.58, 50.96, 45.96, 39.42, 32.06, 30.90, 30.34, 29.36, 27.00, 26.88, 24.34, 20.74, 20.32, 19.17, 10.85. HRMS(ESI) *m*/*z* [M + H]^+^: calcd for C_33_H_37_N_3_O_3_: 524.2868, found 524.2875.

6-bromo-2-oxo-N-(5-((4,11,11-trimethyl-1,2,3,4-tetrahydro-1,4-methanoacridin-9-yl)amino)pentyl)-2H-chromene-3-carboxamide (**7c**): Intermediate **6a** was treated with intermediate **4c** according to general procedure to obtain the desired product **7c** as a yellow solid (38%). mp: 143–145 °C. ^1^H NMR (500 MHz, CDCl_3_) δ 8.82 (s, 1H), 8.80 (s, 1H), 8.06–7.96 (m, 2H), 7.84–7.72 (m, 2H), 7.56–7.49 (m, 1H), 7.36 (d, *J* = 7.8 Hz, 1H), 7.30 (d, *J* = 8.9 Hz, 1H), 4.62 (bs, 1H), 3.59 (d, *J* = 6.4 Hz, 1H), 3.55–3.46 (m, 3H), 3.30–3.25 (m, 1H), 2.20–2.12 (m, 1H), 1.92–1.85 (m, 1H), 1.85–1.66 (m, 4H), 1.65–1.53 (m, 2H), 1.31 (s, 3H), 1.27 (d, *J* = 7.4 Hz, 2H), 1.02 (s, 3H), 0.66 (s, 3H).^13^C NMR (125 MHz, CDCl_3_) δ 173.20, 160.83, 153.07, 147.27, 146.82, 142.22, 136.63, 131.74, 129.43, 127.37, 123.74, 120.06, 119.98, 119.51, 119.38, 118.26, 117.97, 117.88,54.30, 53.53, 50.92, 46.04, 39.84, 32.07, 30.69, 29.17, 28.99, 26.97, 26.80, 20.31, 19.18, 10.84. HRMS (ESI) *m*/*z* [M + H]^+^: calcd for C_32_H_34_BrN_3_O_3_: 588.1862, found 588.1848.

2-oxo-6-(trifluoromethoxy)-N-(5-((4,11,11-trimethyl-1,2,3,4-tetrahydro-1,4-methanoacridin-9-yl)amino)pentyl)-2H-chromene-3-carboxamide(**7d**): Intermediate **6a** was treated with intermediate **4d** according to general procedure to obtain the desired product **7d** as a faint yellow solid (42%). mp: 105–107 °C. ^1^H NMR (500 MHz, CDCl_3_) δ 8.87 (s, 1H), 8.80 (s, 1H), 7.97 (d, *J* = 8.3 Hz, 1H), 7.76–7.71 (m, 1H), 7.56–7.48 (m, 3H), 7.45 (d, *J* = 8.9 Hz, 1H), 7.37–7.32 (m, 1H), 4.51 (bs, 1H), 3.65–3.56 (m, 1H), 3.55–3.47 (m, 3H), 3.28 (t, *J* = 6.8 Hz, 1H), 2.20–2.12 (m, 1H), 1.91–1.85 (m, 1H), 1.83–1.69 (m, 4H), 1.62–1.55 (m, 2H), 1.38 (s, 3H), 1.33 (d, *J* = 8.6 Hz, 2H), 1.03 (s, 3H), 0.65 (s, 3H).^13^C NMR (125 MHz, CDCl_3_) δ 173.21, 160.92, 160.84, 152.33, 147.19, 145.62, 142.27, 129.37, 127.42, 126.93, 123.78, 121.28, 119.98, 119.70, 119.48, 119.23, 118.27, 118.02, 54.34, 53.57, 50.97, 45.94, 39.57, 32.08, 30.91, 30.33, 29.30, 27.01, 24.31, 20.33, 19.19,14.096, 10.86. HRMS(ESI) *m*/*z* [M + H]^+^: calcd for C_33_H_34_F_3_N_3_O_4_: 588.1862, found 588.1883.

6-methoxy-2-oxo-N-(5-((4,11,11-trimethyl-1,2,3,4-tetrahydro-1,4-methanoacridin-9-yl)amino)pentyl)-2H-chromene-3-carboxamide (**7e**): Intermediate **6a** was treated with intermediate **4e** according to general procedure to obtain the desired product **7e** as a yellow solid (40%). mp: 127–129 °C. ^1^H NMR (500 MHz, CDCl_3_) δ 8.91 (s, 1H), 8.83 (s, 1H), 7.98 (d, *J* = 8.3 Hz, 1H), 7.74 (d, *J* = 8.2 Hz, 1H), 7.53 (d, *J* = 11.2, 1H), 7.40–7.34 (m, 1H), 7.33 (d, *J* = 9.1 Hz, 1H), 7.23 (d, *J* = 9.1 Hz, 1H), 7.03 (d, *J* = 2.9 Hz, 1H), 4.55 (bs, 1H), 3.87 (s, 3H), 3.62–3.54 (m, 1H), 3.52–3.45 (m, 3H), 3.27 (d, *J* = 4.0 Hz, 1H), 2.20–2.12 (m, 1H), 1.91–1.82 (m, 1H), 1.78–1.64 (m, 4H), 1.60–1.44 (m, 2H), 1.38 (s, 3H), 1.32 (t, *J* = 7.5 Hz, 2H), 1.02 (s, 3H), 0.64 (s, 3H).^13^C NMR (125 MHz, CDCl_3_) δ 173.142,161.62, 161.56, 156.58, 148.93, 148.06, 129.28, 127.47, 123.82, 122.55, 119.98, 119.52, 118.95, 118.55, 117.93, 117.66, 110.59, 55.90, 54.35, 53.57, 50.97, 45.82, 39.51, 32.07, 30.91, 30.64, 29.30, 26.99, 26.60, 26.43, 20.30, 19.18, 10.85. HRMS(ESI) *m*/*z* [M + H]^+^: calcd for C_33_H_37_N_3_O_4_: 540.2818, found 540.2850.

7-methoxy-2-oxo-N-(5-((4,11,11-trimethyl-1,2,3,4-tetrahydro-1,4-methanoacridin-9-yl)amino)pentyl)-2H-chromene-3-carboxamide (**7f**): Intermediate **6a** was treated with intermediate **4f** according to general procedure to obtain the desired product **7f** as a white solid (38%). mp: 123–125 °C. ^1^H NMR (500 MHz, CDCl_3_) δ 8.83 (s, 1H), 8.82 (s, 1H),7.95 (d, *J* = 8.3, 1H), 7.74 (d, *J* = 7.7 Hz, 1H), 7.55 (d, *J* = 8.7 Hz, 1H), 7.53–7.48 (m, 1H), 7.36–7.31 (m, 1H), 6.92 (dd, *J* = 8.7, 2.4 Hz, 1H), 6.85 (d, *J* = 2.3 Hz, 1H), 4.53 (bs, 1H), 3.90 (s, 3H), 3.63–3.55 (m, 1H), 3.53–3.46 (m, 3H), 3.27 (d, *J* = 4.0 Hz, 1H), 2.19–2.13 (m, 1H), 1.87 (t, *J* = 9.5 Hz, 1H), 1.82–1.74 (m, 2H), 1.73–1.67 (m, 2H), 1.61–1.54 (m, 2H), 1.37 (s, 3H), 1.32 (d, *J* = 8.3 Hz, 2H), 1.02 (s, 3H), 0.64 (s, 3H).^13^C NMR (125 MHz, CDCl_3_) δ 173.21, 164.78, 162.09, 161.87, 156.55, 148.22, 147.28, 142.23, 130.86, 129.40, 127.33, 123.71, 119.99, 119.53, 117.95, 114.69, 113.98, 112.35, 100.24, 55.97, 54.28, 53.51, 50.93, 45.96, 39.32, 32.06, 30.33, 29.41, 26.99, 24.32, 20.31, 19.16, 10.84. HRMS(ESI) *m*/*z* [M + H]^+^: calcd for C_33_H_37_N_3_O_4_: 540.2818, found 540.2847.

6,7-dimethoxy-2-oxo-N-(5-((4,11,11-trimethyl-1,2,3,4-tetrahydro-1,4-methanoacridin-9-yl)amino)pentyl)-2H-chromene-3-carboxamide (**7g**): Intermediate **6a** was treated with intermediate **4g** according to general procedure to obtain the desired product **7g** as a faint yellow solid (43%). mp: 86–88 °C. ^1^H NMR (500 MHz, CDCl_3_) δ 8.89 (s, 1H), 8.83 (s, 1H), 7.96 (d, *J* = 8.0 Hz, 1H), 7.76 (d, *J* = 7.7 Hz, 1H), 7.53–7.48 (m, 1H), 7.39–7.32 (m, 1H), 6.98 (d, *J* = 4.4 Hz, 1H), 6.88 (s, 1H), 4.53 (bs, 1H), 3.99 (s, 3H), 3.94 (s, 3H), 3.63–3.56 (m, 1H), 3.55–3.47 (m, 3H), 3.28 (d, *J* = 4.0 Hz, 1H), 2.20–2.12 (m, 1H), 1.87 (t, *J* = 9.5 Hz, 1H), 1.83–1.68 (m, 4H), 1.62–1.55 (m, 2H), 1.40–1.36 (m, 3H), 1.32 (t, *J* = 8.6 Hz, 2H), 1.04 (s, 3H), 0.65 (s, 3H).^13^C NMR (125 MHz, CDCl_3_) δ 173.26, 162.24, 162.03, 155.13, 151.15, 148.07, 147.17, 142.29, 129.43, 127.34, 123.75, 120.04, 119.59, 118.00, 114.98, 111.50, 108.71, 99.33,56.60, 56.39, 54.32, 53.55, 50.97, 46.01, 39.31, 32.09, 30.91, 30.31, 29.45, 27.02, 24.34, 20.34, 19.19, 10.86. HRMS(ESI) *m*/*z* [M + H]^+^: calcd for C_34_H_39_N_3_O_5_: 571.3002, found 570.2987.

2-oxo-N-(6-((4,11,11-trimethyl-1,2,3,4-tetrahydro-1,4-methanoacridin-9-yl)amino)hexyl)-2H-chromene-3-carboxamide (**8a**): Intermediate **6b** was treated with intermediate **4a** according to general procedure to obtain the desired product **8a** as a yellow solid (28%). mp: 103–105 °C. ^1^H NMR (500 MHz, CDCl_3_) δ 8.90 (s, 1H), 8.87 (s, 1H), 8.03 (d, *J* = 8.2 Hz, 1H), 7.75 (d, *J* = 8.2 Hz, 1H), 7.67 (t, *J* = 7.9 Hz, 2H), 7.52 (t, *J* = 7.5 Hz, 1H), 7.43–7.34 (m, 3H), 7.19 (t, *J* = 7.6 Hz, 1H), 6.88 (t, *J* = 7.5 Hz, 1H), 4.58 (bs, 1H), 3.62 -3.56 (m, 1H), 3.54–3.45 (m, 3H), 3.28 (d, *J* = 4.0 Hz, 1H), 2.22–2.13 (m, 1H), 1.89 (t, *J* = 9.6 Hz, 1H), 1.79–1.64 (m, 4H), 1.59–1.44 (m, 4H), 1.39 (s, 3H), 1.33 (d, *J* = 6.8 Hz, 2H), 1.02 (s, 3H), 0.66 (s, 3H).^13^C NMR (125 MHz, CDCl_3_) δ 172.93,161.52, 161.50, 154.33, 148.30, 134.03, 129.76, 129.40, 127.76, 125.29, 123.99, 119.82,119.53, 118.58, 118.38, 117.77,116.56, 115.65, 54.47, 53.73, 50.98, 45.77, 39.55, 32.04, 30.90,30.62, 29.27, 26.95, 26.60, 26.43, 20.28, 19.14, 10.86. HRMS(ESI) *m*/*z* [M + H]^+^: calcd for C_33_H_37_N_3_O_3_: 524.2868, found 524.2898.

7-methyl-2-oxo-N-(6-((4,11,11-trimethyl-1,2,3,4-tetrahydro-1,4-methanoacridin-9-yl)amino)hexyl)-2H-chromene-3-carboxamide (**8b**): Intermediate **6b** was treated with intermediate **4b** according to general procedure to obtain the desired product **8b** as a yellow solid (36%). mp: 131–133 °C. ^1^H NMR (500 MHz, CDCl_3_) δ 8.89 (s, 1H), 8.85 (s, 1H), 8.07 (d, *J* = 8.0 Hz, 1H), 7.79 (d, *J* = 8.3 Hz, 1H), 7.54 (t, *J* = 7.6 Hz, 1H), 7.47 (d, *J* = 8.6 Hz, 1H), 7.44 (s, 1H), 7.38 (t, *J* = 7.6 Hz, 1H), 7.30 (d, *J* = 8.5 Hz, 1H), 4.79 (bs, 1H), 3.67–3.56 (m, 1H), 3.55–3.43 (m, 3H), 3.28 (d, J = 4.0 Hz, 1H), 2.44 (s, 3H), 2.21–2.13 (m, 1H), 1.89 (t, *J* = 9.7 Hz, 1H), 1.78–1.65 (m, 4H), 1.59–1.44 (m, 4H), 1.40 (s, 3H), 1.34 (d, *J* = 9.0 Hz, 2H), 1.03 (s, 3H), 0.66 (s, 3H).^13^C NMR (125 MHz, CDCl_3_) δ 172.52,161.70, 161.65, 152.53, 148.26, 143.08, 135.20, 135.16, 133.85, 129.33, 128.50, 127.84, 124.06, 119.75, 118.34, 118.19, 117.62, 116.26, 54.57, 53.71, 51.03, 45.69, 39.47, 32.02, 30.59, 29.28, 26.93, 26.56, 26.38, 25.96, 20.73, 20.25, 19.12, 10.94. HRMS(ESI) *m*/*z* [M + H]^+^: calcd for C_34_H_39_N_3_O_3_: 538.3025, found 538.3056.

7-bromo-2-oxo-N-(6-((4,11,11-trimethyl-1,2,3,4-tetrahydro-1,4-methanoacridin-9-yl)amino)hexyl)-2H-chromene-3-carboxamide (**8c**): Intermediate **6b** was treated with intermediate **4c** according to general procedure to obtain the desired product **8c** as a yellow solid (45%). mp: 130–132 °C.^1^H NMR (500 MHz, CDCl_3_) δ 8.88 (s, 1H),8.86 (s, 1H), 7.98 (d, *J* = 8.2 Hz, 1H), 7.72 (d, *J* = 8.2 Hz, 1H), 7.52 (t, *J* = 7.4 Hz, 1H), 7.49–7.41 (m, 2H), 7.36 (t, *J* = 7.4 Hz, 1H), 7.28 (d, *J* = 2.9 Hz, 1H), 4.55 (bs, 1H), 3.62–3.53 (m, 1H), 3.51–3.41 (m, 3H), 3.28 (d, *J* = 3.6 Hz, 1H), 1.94–1.83 (m, 1H), 1.77–1.60 (m, 4H), 1.55–1.45 (m, 4H), 1.44–1.42 (m, 1H), 1.38 (s, 3H), 1.36–1.31 (m, 2H), 1.03 (s, 3H), 0.66 (s, 3H).^13^C NMR (125 MHz, CDCl_3_) δ 173.12, 161.71, 161.55, 152.53, 148.21, 135.15, 129.31, 127.44, 123.79, 119.45, 118.37, 118.26, 116.25,54.34, 53.55, 50.93, 46.25, 46.22, 46.02, 39.72, 32.06, 30.68, 29.20, 28.98, 26.96, 26.81, 26.78, 26.40, 26.34, 20.72, 20.29, 19.17, 10.83. HRMS(ESI) *m*/*z* [M + H]^+^: calcd for C_33_H_36_BrN_3_O_3_: 603.1920, found 603.1903.

2-oxo-7-(trifluoromethoxy)-N-(6-((4,11,11-trimethyl-1,2,3,4-tetrahydro-1,4-methanoacridin-9-yl)amino)hexyl)-2H-chromene-3-carboxamide (**8d**): Intermediate **6b** was treated with intermediate **4d** according to general procedure to obtain the desired product **8d** as a yellow solid (37%). mp: 106–108 °C. ^1^H NMR (500 MHz, CDCl_3_) δ 8.87 (s, 1H), 8.78 (t, *J* = 5.3 Hz, 1H), 7.98 (d, *J* = 7.8 Hz, 1H), 7.74 (dd, *J* = 7.1, 6.3 Hz, 1H), 7.58–7.49 (m, 3H), 7.45 (d, *J* = 9.0 Hz, 1H), 7.40–7.34 (m, 1H), 4.55 (bs, 1H), 3.63–3.56 (m, 1H), 3.54–3.45 (m, 3H), 3.28 (d, *J* = 4.0 Hz, 1H), 1.89 (t, *J* = 9.5 Hz, 1H), 1.84–1.80 (m, 1H), 1.79–1.65 (m, 4H), 1.60–1.45 (m, 4H), 1.39 (s, 3H), 1.34 (d, *J* = 8.6 Hz, 2H), 1.04 (s, 3H), 0.66 (s, 3H). ^13^C NMR (125 MHz, CDCl_3_) δ 173.14, 160.81, 160.78, 152.31, 147.10, 145.59, 142.32, 129.32, 127.46, 126.87, 123.82, 121.27, 119.97, 119.75, 119.44, 119.23, 118.25, 117.97, 54.35, 53.57, 50.96, 46.27, 46.24, 45.86, 39.67, 32.07, 30.64, 29.24, 26.98, 26.64, 26.48, 20.30, 19.18, 10.83. HRMS(ESI) *m*/*z* [M + H]^+^: calcd for C_34_H_36_F_3_N_3_O_4_: 608.2691, found 608.2722.

6-methoxy-2-oxo-N-(6-((4,11,11-trimethyl-1,2,3,4-tetrahydro-1,4-methanoacridin-9-yl)amino)hexyl)-2H-chromene-3-carboxamide (**8e**): Intermediate **6b** was treated with intermediate **4e** according to general procedure to obtain the desired product **8e** as a yellow solid (38%). mp: 107–109 °C. ^1^H NMR (500 MHz, CDCl_3_) δ 8.91 (s, 1H), 8.84 (s, 1H), 7.98 (d, *J* = 8.3 Hz, 1H), 7.75 (d, *J* = 8.3 Hz, 1H), 7.52 (t, *J* = 7.6 Hz, 1H), 7.37 (t, *J* = 7.6 Hz, 1H), 7.33 (d, *J* = 9.1 Hz, 1H), 7.23 (dd, *J* = 9.1, 2.8 Hz, 1H), 7.03 (d, *J* = 2.7 Hz, 1H), 4.51 (bs, 1H), 3.86 (s, 3H), 3.64–3.54 (m, 1H), 3.54–3.44 (m, 3H), 3.27 (d, *J* = 3.9 Hz, 1H), 2.22–2.10 (m, 1H), 1.92–1.83 (m, 1H), 1.78–1.63 (m, 4H), 1.60–1.43 (m, 4H), 1.37 (s, 3H), 1.35–1.30 (m, 2H), 1.02 (s, 3H), 0.65 (s, 3H). ^13^C NMR (125 MHz, CDCl_3_) δ 173.09, 161.60, 161.54, 156.56, 148.91, 148.05, 142.40, 129.21, 127.47, 123.82, 122.53, 119.96, 119.53, 118.92, 118.52, 117.88, 117.64, 110.57, 55.88, 54.34, 53.55, 50.95, 45.78, 39.48, 32.05, 30.61, 29.28, 26.96, 26.57, 26.40, 20.27, 19.15, 10.82, 8.19. HRMS(ESI) *m*/*z* [M + H]^+^: calcd for C_34_H_39_N_3_O_4_: 554.2974, found 554.2987.

7-methoxy-2-oxo-N-(6-((4,11,11-trimethyl-1,2,3,4-tetrahydro-1,4-methanoacridin-9-yl)amino)hexyl)-2H-chromene-3-carboxamide (**8f**): Intermediate **6b** was treated with intermediate **4f** according to general procedure to obtain the desired product **8f** as a white solid (39%). mp: 103–105 °C. ^1^H NMR (500 MHz, CDCl_3_) δ 8.84 (s, 1H), 8.80 (s, 1H), 7.98 (d, *J* = 8.3 Hz, 1H), 7.74 (dd, *J* = 8.3, 0.8 Hz, 1H), 7.57 (d, *J* = 8.7 Hz, 1H), 7.53 (t, *J* = 7.6 Hz, 1H), 7.37 (t, *J* = 7.6 Hz, 1H), 6.94 (dd, *J* = 8.7, 2.4 Hz, 1H), 6.86 (d, *J* = 2.3 Hz, 1H), 4.51 (bs, 1H), 3.92 (s, 3H), 3.65–3.55 (m, 1H), 3.52–3.46 (m, 3H), 3.28 (d, *J* = 4.0 Hz, 1H), 1.88 (t, *J* = 9.5 Hz, 1H), 1.78–1.64 (m, 4H), 1.60–1.46 (m, 5H), 1.43 (s, 1H), 1.39 (s, 3H), 1.34 (d, *J* = 8.7 Hz, 2H), 1.03 (s, 3H), 0.66 (s, 3H). ^13^C NMR (125 MHz, CDCl_3_) δ 173.19, 164.77, 161.99, 161.89, 156.57, 148.18, 142.29, 130.87, 129.40, 127.40, 123.78, 120.01, 119.47, 117.97, 114.79, 113.98, 112.39, 100.26, 55.99, 54.33, 53.55, 50.95, 45.89, 39.47, 32.09, 30.67, 29.35, 26.99, 26.88, 26.66, 26.50, 20.31, 19.19, 10.85. HRMS(ESI) *m*/*z* [M + H]^+^: calcd for C_34_H_39_N_3_O_4_: 554.2974, found 554.3013.

6,7-dimethoxy-2-oxo-N-(6-((4,11,11-trimethyl-1,2,3,4-tetrahydro-1,4-methanoacridin-9-yl)amino)hexyl)-2H-chromene-3-carboxamide (**8g**): Intermediate **6b** was treated with intermediate **4g** according to general procedure to obtain the desired product **8g** as a faint yellow solid (41%). mp: 96–98 °C. ^1^H NMR (500 MHz, CDCl_3_) δ 8.86 (d, *J* = 5.3 Hz, 1H), 8.79 (s, 1H), 7.99 (d, *J* = 8.3 Hz, 1H), 7.76 (d, *J* = 8.2 Hz, 1H), 7.53 (t, *J* = 7.6 Hz, 1H), 7.37 (dd, *J* = 8.1, 7.0 Hz, 1H), 6.96 (s, 1H), 6.88 (s, 1H), 4.23 (bs, 1H), 3.99 (s, 3H), 3.95 (s, 3H), 3.58 (dd, *J* = 12.5, 6.3 Hz, 1H), 3.53–3.44 (m, 3H), 3.27 (d, *J* = 3.9 Hz, 1H), 1.87 (t, *J* = 9.5 Hz, 1H), 1.81 (d, *J* = 6.4 Hz, 1H), 1.78–1.63 (m, 4H), 1.60–1.44 (m, 4H), 1.37 (s, 3H), 1.33 (d, *J* = 9.0 Hz, 2H), 1.03 (s, 3H), 0.64 (s, 3H). ^13^C NMR (125 MHz, CDCl_3_) δ 173.12, 162.12, 162.00, 155.08, 151.11, 148.01, 147.14, 142.44, 129.21, 127.49, 123.84, 119.99, 119.56, 117.91, 115.00, 111.48, 108.71, 99.31, 56.59, 56.39, 54.36, 53.58, 50.97, 46.26, 45.78, 39.39, 32.06, 30.63, 29.36, 26.98, 26.56, 26.40, 20.29, 19.17, 10.86. HRMS(ESI) *m*/*z* [M + H]^+^: calcd for C_35_H_41_N_3_O_5_: 584.3080, found 584.3113.

2-oxo-N-(7-((4,11,11-trimethyl-1,2,3,4-tetrahydro-1,4-methanoacridin-9-yl)amino)heptyl)-2H-chromene-3-carboxamide (**9a**): Intermediate **6c** was treated with intermediate **4a** according to general procedure to obtain the desired product **9a** as a yellow solid (42%). mp: 99–101 °C. ^1^H NMR (500 MHz, CDCl_3_) δ 8.90 (s, 1H), 8.83 (s, 1H), 8.00 (d, *J* = 7.1 Hz, 1H), 7.79–7.70 (m, 1H), 7.70–7.60 (m, 1H), 7.48 (m, 1H), 7.43–7.29 (m, 3H), 4.79 (bs, 1H), 3.63–3.52 (m, 1H), 3.50–3.43 (m, 3H), 3.27 (s, 1H), 1.92–1.80 (m, 1H), 1.75–1.60 (m, 4H), 1.53–1.40 (m, 6H), 1.41–1.36 (m, 4H), 1.33 (s, 3H), 1.27–1.22 (m, 2H), 1.01 (s, 3H), 0.65 (s, 3H).^13^C NMR (125 MHz, CDCl_3_) δ172.23, 161.49, 161.39, 154.32, 148.22, 133.97, 129.73, 127.76, 125.26, 123.99, 119.75, 119.63, 118.59, 118.44, 116.55, 60.34, 58.43, 54.53, 53.67, 50.99, 45.92,39.75, 32.02, 30.65, 29.19, 28.97, 26.92, 26.80, 26.76, 21.01, 20.26, 19.13, 14.15, 10.83. HRMS(ESI) *m*/*z* [M + H]^+^: calcd for C_34_H_39_N_3_O_3_: 538.3025, found 538.3057.

6-methyl-2-oxo-N-(7-((4,11,11-trimethyl-1,2,3,4-tetrahydro-1,4-methanoacridin-9-yl)amino)heptyl)-2H-chromene-3-carboxamide (**9b**): Intermediate **6c** was treated with intermediate **4b** according to general procedure to obtain the desired product **9b** as a faint yellow solid (37%). mp: 113–114 °C. ^1^H NMR (500 MHz, CDCl_3_) δ 8.87 (s, 1H), 8.86 (s, 1H), 7.99 (d, *J* = 8.3 Hz, 1H), 7.72 (d, *J* = 8.3 Hz, 1H), 7.52 (t, *J* = 7.6 Hz, 1H), 7.48–7.44 (m, 2H), 7.41–7.33 (m, 1H), 7.29 (d, *J* = 8.3 Hz, 1H), 4.59 (bs, 1H), 3.61–3.53 (m, 1H), 3.52–3.42 (m, 3H), 3.28 (d, *J* = 4.0 Hz, 1H), 2.44 (s, 3H), 1.92–1.85 (m, 1H), 1.85–1.78 (m, 1H), 1.76–1.62 (m, 4H), 1.54–1.42 (m, 6H), 1.39 (s, 3H), 1.34 (d, *J* = 8.7 Hz, 2H), 1.02 (s, 3H), 0.66 (s, 3H). ^13^C NMR (125 MHz, CDCl_3_) δ173.01, 161.70, 161.55, 152.52, 148.21, 143.08, 135.15, 129.31, 128.24, 127.55, 123.86, 119.87, 119.49, 118.37, 118.25, 117.82, 116.25, 54.40, 53.60, 50.95, 46.00, 39.73, 32.05, 30.67, 29.20, 28.98, 26.96, 26.82, 26.78, 25.96, 24.26, 20.73, 20.29, 19.16, 10.83. HRMS(ESI) *m*/*z* [M + H]^+^: calcd for C_35_H_41_N_3_O_3_: 552.3181, found 552.3220.

6-bromo-2-oxo-N-(7-((4,11,11-trimethyl-1,2,3,4-tetrahydro-1,4-methanoacridin-9-yl)amino)heptyl)-2H-chromene-3-carboxamide (**9c**): Intermediate **6c** was treated with intermediate **4c** according to general procedure to obtain the desired product **9c** as a faint yellow oil (30%). ^1^H NMR (500 MHz, CDCl_3_) δ 8.82 (s, 1H), 8.75 (s, 1H), 7.97 (d, *J* = 8.3 Hz, 1H), 7.80 (s, 1H), 7.76–7.70 (m, 1H), 7.70 (d, *J* = 8.5 Hz, 1H), 7.52 (t, *J* = 7.6 Hz, 1H), 7.35 (t, *J* = 7.6 Hz, 1H), 7.30–7.26 (m, 1H), 4.47 (s, 1H), 3.56 (d, *J* = 6.5 Hz, 1H), 3.46 (dd, *J* = 13.1, 6.7 Hz, 3H), 3.26 (d, *J* = 3.9 Hz, 1H), 2.21–2.11 (m, 2H), 1.87 (t, *J* = 9.5 Hz, 1H), 1.74–1.62 (m, 4H), 1.51–1.40 (m, 6H), 1.37 (s, 3H), 1.33 (d, *J* = 8.7 Hz, 2H), 1.02 (s, 3H), 0.64 (s, 3H). ^13^C NMR (125 MHz, CDCl_3_) δ 173.23, 160.85, 153.10, 147.29, 146.84, 142.23, 136.65, 131.76, 129.46, 127.39, 123.76, 120.08, 120.00, 119.53, 119.39, 118.29, 118.01, 117.91, 54.32, 53.55, 50.94, 46.07, 39.86, 32.09, 30.71, 29.19, 29.01, 26.99, 26.84, 26.81,26.498, 20.32, 19.20, 10.86. HRMS(ESI) *m*/*z* [M + H]^+^: calcd for C_34_H_38_BrN_3_O_3_:618.2154, found 618.2177.

2-oxo-6-(trifluoromethoxy)-N-(7-((4,11,11-trimethyl-1,2,3,4-tetrahydro-1,4-methanoacridin-9-yl)amino)heptyl)-2H-chromene-3-carboxamide (**9d**): Intermediate **6c** was treated with intermediate **4d** according to general procedure to obtain the desired product **9d** as a faint yellow solid (33%). mp: 89–91 °C. ^1^H NMR (500 MHz, CDCl_3_) δ 8.87 (s, 1H), 8.75 (s, 1H), 8.00 (d, *J* = 9.5 Hz, 1H), 7.71 (d, *J* = 8.5 Hz, 1H), 7.58–7.48 (m, 3H), 7.44 (d, *J* = 9.0 Hz, 1H), 7.40–7.32 (m, 1H), 4.56 (bs, 1H), 3.62–3.53 (m, 1H), 3.52–3.42 (m, 3H), 3.29–3.24 (m, 1H), 1.96–1.90 (m, 1H), 1.89–1.84 (m, 1H), 1.76–1.61 (m, 4H), 1.54–1.41 (m, 6H), 1.38 (s, 3H), 1.33 (d, *J* = 8.5 Hz, 2H), 1.02 (s, 3H), 0.65 (s, 3H). ^13^C NMR (125 MHz, CDCl_3_) δ172.97, 160.82, 160.72, 152.32, 147.08, 127.56, 126.85, 123.87, 121.26, 119.89, 119.80, 119.45, 119.24, 118.24, 117.89, 109.99, 54.41, 53.61, 50.96, 47.02, 46.98, 46.00, 39.87, 32.05, 30.68, 29.17, 28.99, 26.96, 26.82, 26.79, 26.34, 26.27, 20.29, 19.16, 10.82. HRMS(ESI) *m*/*z* [M + H]^+^: calcd for C_35_H_38_F_3_N_3_O_4_: 622.2848, found 622.2868.

6-methoxy-2-oxo-N-(7-((4,11,11-trimethyl-1,2,3,4-tetrahydro-1,4-methanoacridin-9-yl)amino)heptyl)-2H-chromene-3-carboxamide (**9e**): Intermediate **6c** was treated with intermediate **4e** according to general procedure to obtain the desired product **9e** as a faint yellow solid (46%). mp: 104–106 °C. ^1^H NMR (500 MHz, CDCl_3_) δ 8.88 (s, 1H), 8.84 (s, 1H), 8.11 (d, *J* = 7.7 Hz, 1H), 7.84 (dd, *J* = 21.8, 8.3 Hz, 1H), 7.55–7.46 (m, 1H), 7.39–7.28 (m, 2H), 7.25–7.19 (m, 1H), 7.08–7.01 (m, 1H), 4.11 (bs, 1H), 3.85 (s, 3H), 3.63–3.55 (m, 1H), 3.54–3.41 (m, 3H), 3.29–3.24 (m, 1H), 1.94–1.83 (m, 1H), 1.77–1.57 (m, 4H), 1.54–1.38 (m, 7H), 1.31 (s, 3H), 1.30–1.25 (m, 2H), 1.01 (s, 3H), 0.65 (s, 3H). ^13^C NMR (125 MHz, CDCl_3_) δ171.92, 161.58, 161.46, 156.54, 148.88, 148.00, 128.20, 124.27, 122.48, 120.07, 119.49, 118.91, 118.53, 117.61, 117.22, 110.59,58.45, 55.86, 54.78, 53.82, 51.08, 45.72,43.17, 39.70, 31.95, 30.58, 29.15, 28.91, 26.85, 26.76, 26.69, 20.19, 19.05, 14.05, 10.87. HRMS(ESI) *m*/*z* [M + H]^+^: calcd for C_35_H_41_N_3_O_4_: 568.3131, found 568.3158.

7-methoxy-2-oxo-N-(7-((4,11,11-trimethyl-1,2,3,4-tetrahydro-1,4-methanoacridin-9-yl)amino)heptyl)-2H-chromene-3-carboxamide (**9f**): Intermediate **6c** was treated with intermediate **4f** according to general procedure to obtain the desired product **9f** as a white solid (38%). mp: 99–101 °C. ^1^H NMR (500 MHz, CDCl_3_) δ 8.83 (s, 1H), 8.78 (s, 1H), 7.98 (d, *J* = 8.3 Hz, 1H), 7.70 (d, *J* = 9.0 Hz, 1H), 7.56 (dd, *J* = 8.7, 4.0 Hz, 1H), 7.52 (t, *J* = 7.5 Hz, 1H), 7.35 (t, *J* = 7.6 Hz, 1H), 6.93 (dt, *J* = 8.7, 2.9 Hz, 1H), 6.85 (d, J = 2.6 Hz, 1H), 4.50 (bs, 1H), 3.90 (s, 3H), 3.60–3.52 (m, 1H), 3.51–3.42 (m, 3H), 3.27 (d, *J* = 3.9 Hz, 1H), 2.21–2.12 (m, 1H), 1.90–1.83 (m, 1H), 1.76–1.60 (m, 4H), 1.53–1.40 (m, 6H), 1.37 (s, 3H), 1.32 (t, *J* = 8.8 Hz, 2H), 1.02 (s, 3H), 0.65 (s, 3H). ^13^C NMR (125 MHz, CDCl_3_) δ 173.13, 161.62, 161.56, 156.58, 148.93, 148.06, 147.11, 142.38, 129.28, 127.47, 123.82, 122.55, 119.98, 119.52, 118.95, 118.55, 117.93, 117.66, 110.59, 55.90, 54.35, 53.57, 50.97, 45.82, 39.51, 32.07, 30.91, 30.64, 29.30, 26.99, 26.60, 26.43, 20.30, 19.18, 10.85. HRMS(ESI) *m*/*z* [M + H]^+^: calcd for C_35_H_41_N_3_O_4_: 568.3131, found 568.3156.

6,7-dimethoxy-2-oxo-N-(7-((4,11,11-trimethyl-1,2,3,4-tetrahydro-1,4-methanoacridin-9-yl)amino)heptyl)-2H-chromene-3-carboxamide (**9g**): Intermediate **6c** was treated with intermediate **4g** according to general procedure to obtain the desired product **9g** as a yellow solid (36%). mp: 83–86 °C. ^1^H NMR (500 MHz, CDCl_3_) δ 8.83 (s, 1H), 8.80 (s, 1H), 7.97 (dd, *J* = 8.0, 2.8 Hz, 1H), 7.71 (d, *J* = 8.3 Hz, 1H), 7.57–7.47 (m, 1H), 7.40–7.30 (m, 1H), 6.96 (d, *J* = 7.5 Hz, 1H), 6.86 (d, *J* = 6.6 Hz, 1H), 4.52 (bs, 1H), 4.00 (s, 3H), 3.92 (s, 3H), 3.63–3.52 (m, 1H), 3.51–3.39 (m, 3H), 3.29–3.23 (m, 1H), 1.91–1.83 (m, 1H), 1.75–1.59 (m, 4H), 1.55–1.39 (m, 7H), 1.37 (s, 3H), 1.32 (dd, *J* = 8.4, 4.0 Hz, 2H), 1.01 (s, 3H), 0.64 (s, 3H). ^13^C NMR (125 MHz, CDCl_3_) δ 173.16, 162.02, 155.06, 151.10, 147.95, 147.13, 142.35, 129.36, 127.42, 123.78, 119.98, 119.46, 117.95, 115.07, 111.49, 108.70, 99.32, 56.59, 56.38, 54.34, 53.56, 50.95, 46.25, 46.05, 39.66, 32.09, 30.69, 29.26, 28.99, 26.99, 26.69, 26.43, 26.36, 20.32, 19.19, 10.87. HRMS(ESI) *m*/*z* [M + H]^+^: calcd for C_36_H_43_N_3_O_5_: 598.3236, found 598.3258.

### 4.5. Biological Activity

#### 4.5.1. In Vitro Inhibition Studies on AChE

Acetylcholinesterase (AChE, E.C. 3.1.1.7, from electric eel) and Acetylthiocholine Iodide (ATCI, CAS 69–78-3) were purchased from Sigma Aldrich. In addition, 5,5′-Dithiobis (2-nitrobenzoic acid) (Ellman’s reagent, DTNB) was purchased from Yuanye Biology. Tacrine (THA), Galanthamine (GAL) and synthesized hybrids were dissolved in DMSO and then diluted in 0.1M KH_2_PO_4_/K_2_HPO_4_ buffer (PH 7.4) to provide a final concentration range. The final concentration of DMSO is below 1%.

All the assay was conducted under 0.1M KH_2_PO_4_/K_2_HPO_4_ buffer (PH 7.4), using a Molecule Devices SpectraMax M2 Multiscan Spectrum. Enzyme solutions were prepared to provide 10 units/mL. The assay medium contained phosphate buffer, PH 7.4 (80 µL), 50 µL of AChE and 20 µL of 0.2 µM inhibitors (THA, GLA, coumarin derivatives). The mixture was then incubated at 37 °C for 10 min. Afterward, 40 µL of 15 mM DTNB and 10 µL of 3 mM ATCI in 0.1M KH_2_PO_4_/K_2_HPO_4_ buffer were added and incubated for an additional 4 min at 37 °C. Absorption was subsequently measured using UV spectrometer at a wavelength of 412 nm. The reaction without inhibitor was treated as blank and each experiment was triplicated. The inhibitory activity was calculated using the following equation:

Inhibition % = [(blank absorption − sample absorption)/blank absorption] × 100%

#### 4.5.2. Enzymatic Kinetics of AChE Inhibition

Kinetic characterization was performed using Dixon’s method [27,28]. Plots of 1/velocity versus inhibitors’ concentration were constructed at different final concentrations of the substrate ACTI (10 μM and 5 μM). Eight final concentrations (0.01 μM, 0.05 μM, 0.1 μM, 0.2 μM, 0.4 μM, 0.6 μM, 0.8 μM, 1 μM) of inhibitor **7b**, **7e**, **7f**, **7g**, THA and GAL were used in the studies. Assay medium contained DTNB solution 15 mM (40 µL), AChE solution 10 units/mL (50 µL), substrate solutions (10 µL) and various concentrations of tested compounds (20 µL). Changes in absorbance were detected at 412 nm at room temperature at the time interval (4 min). Results were computed and presented on the Lineweaver–Burk plot to determine the K_i_ value and to distinguish the type of enzyme inhibition.

### 4.6. Molecular Docking, MD Simulation and Binding Energy Calculation

The structures of the compounds were established by the software SYBYL. The structure of AChE was extracted from its complex crystal structure with an inhibitor in the Protein Data Bank (PDB) database (PDB ID 6XYU). The AutoDock 4.2 program was used to dock the compounds into the same binding site as in the crystal structure. The grid size was set to 40 × 55 × 40 Å, and the grid space was set to 0.375 Å. The other parameters were set as default values. Among a set of 100 candidates of dock poses, the best candidate was selected according to the interaction energy and the comparison of the conformation with the crystal structure.

All of the complex structures derived from the molecular docking were used as starting structures for molecular dynamics simulation by the Amber 16 program. The Amber ff14 force field was used for amino acids, and the General Amber force field was used for ligands. The bcc charges were used as the atomic charges for ligands. There are two stages for energy minimization: (1) In the first step, the backbone of AChE and the ligands were constrained; (2) in the second step, both AChE and the ligands were relaxed. Energy minimization was executed using the steepest descent method for the first 2500 cycles and the conjugated gradient method for the subsequent 2500 cycles. Each system was gradually heated from 0 to 300 K during a period of 100 ps in the NVT ensemble, followed by 500 ps equilibration simulation in the NTP (T = 300 K and *p* = 1 atm) ensemble. Finally, 50 ns production MD simulation was performed. The time step was set to 2 fs, and the snapshots were collected at an interval of 50 ps. The last 100 frames were used for the binding free energy calculation. The binding energy of the protein–ligand complex was calculated using the MM/GBSA module.

## Figures and Tables

**Figure 1 molecules-27-02142-f001:**
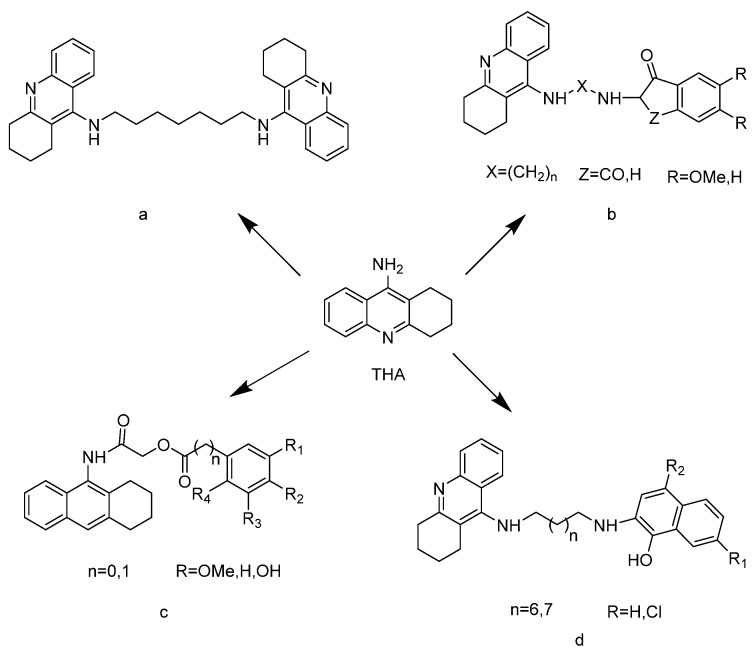
Structure of THA derivatives.

**Figure 2 molecules-27-02142-f002:**
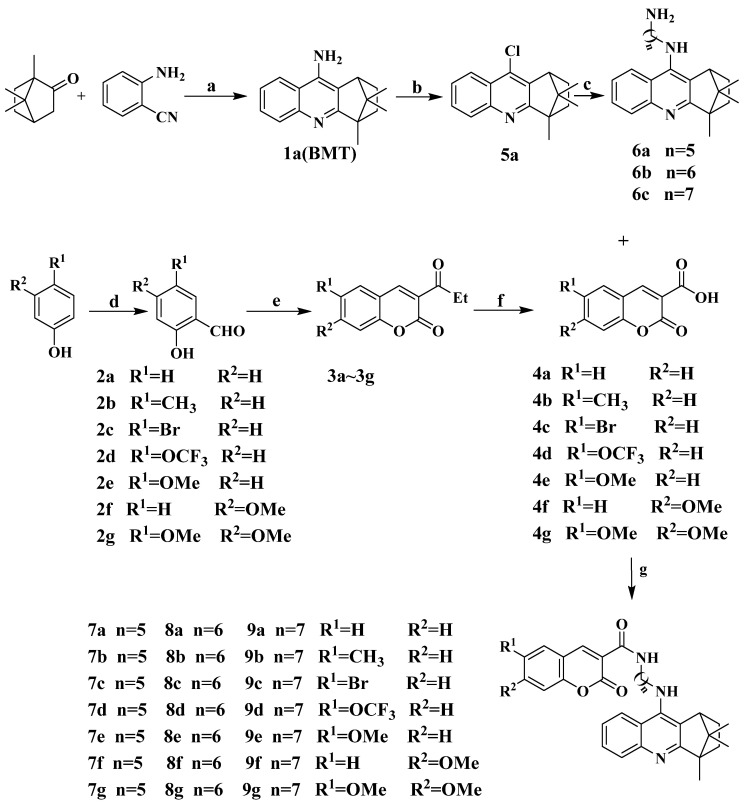
Synthesis routes of target product coumarin–BMT hybrid. Reagents and conditions: (**a**) AlCl_3_, Toluene, NaOH, 140 °C, reflux. (**b**) HCl, NaNO_2_, SnCl_2_; (**c**) phenol, NaI, N_2_, 180 °C, reflux; (**d**) (CH_2_O)_n_, MgCl_2_, CH_3_CN, Et_3_N,95 °C, reflux; (**e**) CH_2_(CO_2_C_2_H_5_)_2_, HAC, EtOH, 95 °C, reflux; (**f**) EtOH, NaOH, 95 °C, reflux. (**g**) PyBOP, Et_3_N, CH_2_Cl_2_, r.t.

**Figure 3 molecules-27-02142-f003:**
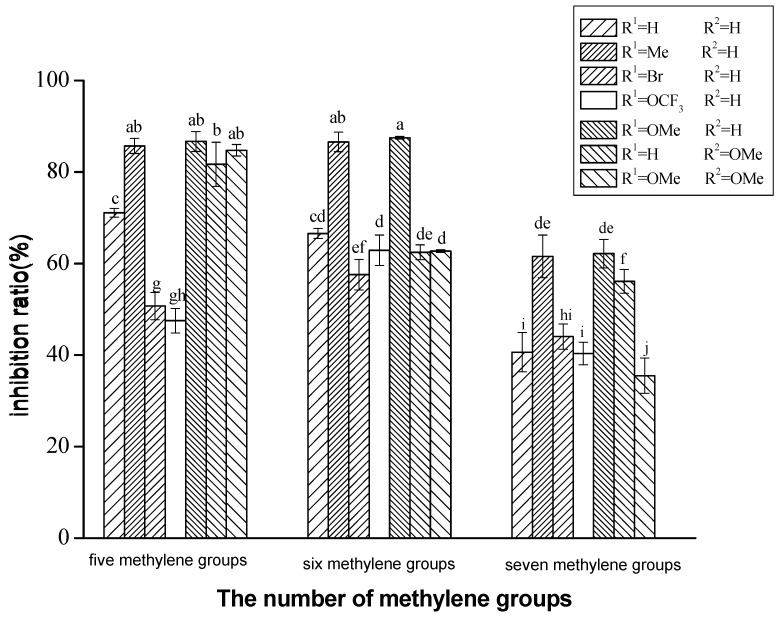
Inhibitory activity of coumarin–BMT hybrids. Note: Data are normalized as a percentage of control, and are expressed as the means ± SEM of at least three independent experiments, and the measurements were carried out in the presence of 2 μM compounds. There are no identical letters on the superscript of the data, showing that they have statistically significant difference (*p* ≤ 0.05) between the two groups. Any letter being the same on the superscript of the data shows that they have no difference (*p* > 0.05) between the two groups. Statistical analysis was performed with ANOVA followed by Duncan’s multiple range test (DMRT).

**Figure 4 molecules-27-02142-f004:**
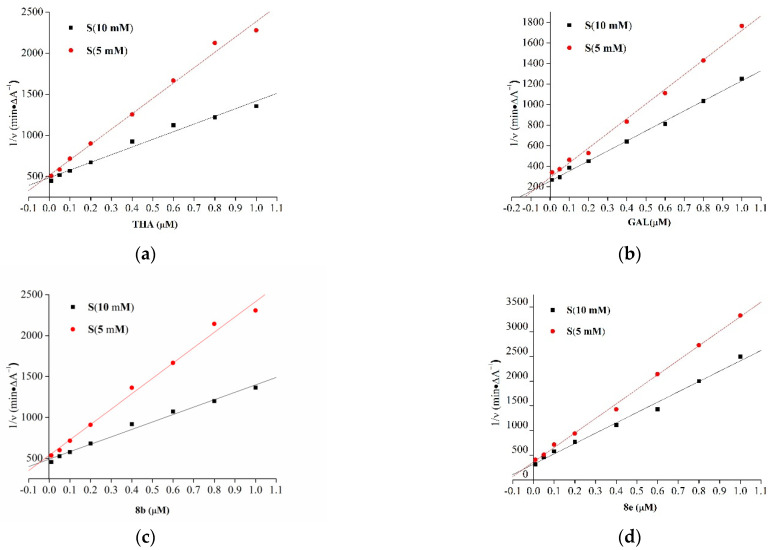
Kinetic parameters (K*_i_*) of AChE inhibition by two high-activity Coumarin–BMT hybrids (**8b** and **8e**) and positive control THA and GAL. Lineweaver-Burk reciprocal plots of the AChE initial velocity at substrate concentrations (5 mM and 10 mM) in the presence of (**a**) THA, (**b**) GAL, (**c**) **8b** and (**d**) **8e**.

**Figure 5 molecules-27-02142-f005:**
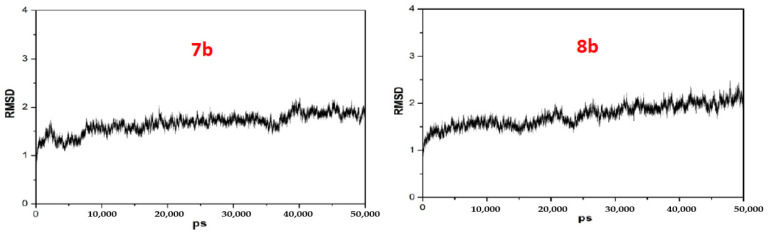
The RMSD value of compound **7b** and **8b**.

**Figure 6 molecules-27-02142-f006:**
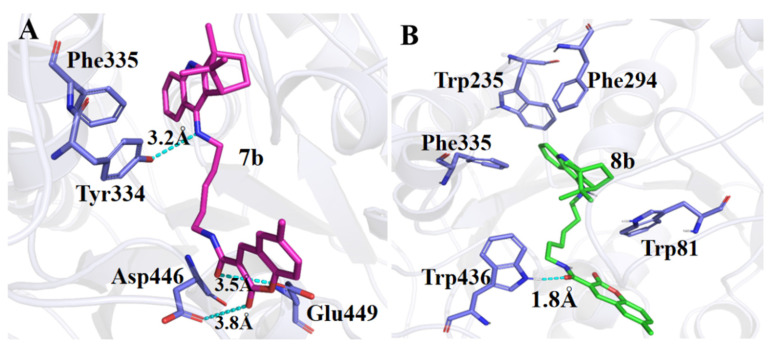
Binding mode of AchE (white cartoon) with the compounds **7b** (**A**) and **8****b** (**B**). The blue dotted lines represent H-bonds. The interaction residues are shown as purple sticks, the compound **7b** as a red salmon stick and the compound **8b** as a green salmon stick.

**Figure 7 molecules-27-02142-f007:**
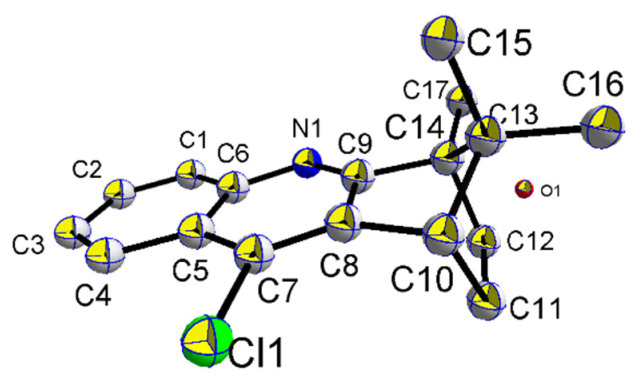
Crystal structure of complex **5a** with 50% thermal ellipsoids. Hydrogen atoms are omitted for clarity.

**Table 1 molecules-27-02142-t001:** Inhibitory activity of Coumarin–BMT hybrids and positive controls against AChE.

Compounds	R^1^ (Position 6) ^[a]^	R^2^ (Position 7) ^[a]^	*n*	Inhibition Ratio (%) ^[b]^
**7a**	H	H	5	71.11 ± 0.94 ^d^
**7b**	Me	H	5	85.71 ± 1.66 ^bc^
**7c**	Br	H	5	50.72 ± 2.98 ^h^
**7d**	OCF_3_	H	5	47.51 ± 2.65 ^hi^
**7e**	OMe	H	5	86.68 ± 2.18 ^b^
**7f**	H	OMe	5	81.70 ± 4.82 ^c^
**7g**	OMe	OMe	5	84.75 ± 1.27 ^bc^
**8a**	H	H	6	66.57 ± 1.09 ^de^
**8b**	Me	H	6	86.60 ± 2.13 ^b^
**8c**	Br	H	6	57.58 ± 3.30 ^fg^
**8d**	OCF_3_	H	6	62.92 ± 3.32 ^e^
**8e**	OMe	H	6	87.48 ± 0.27 ^b^
**8f**	H	OMe	6	62.44 ± 1.63 ^e^
**8g**	OMe	OMe	6	62.74 ± 0.27 ^e^
**9a**	H	H	7	40.65 ± 4.28 ^j^
**9b**	Me	H	7	61.56 ± 4.65 ^ef^
**9c**	Br	H	7	44.05 ± 2.74 ^ij^
**9d**	OCF_3_	H	7	40.35 ± 2.45 ^j^
**9e**	OMe	H	7	62.15 ± 3.14 ^ef^
**9f**	H	OMe	7	56.11 ± 2.60 ^g^
**9g**	OMe	OMe	7	35.49 ± 3.86 ^k^
**THA**	–	–	–	95.75 ± 1.45 ^a^
**GAL**	–	–	–	69.23 ± 2.34 ^d^

Note: ^[a]^ The general structure of Coumarin–BMT hybrids is included in Figure 2. ^[b]^ Data are normalized as a percentage of control and are expressed as the means ± SEM of at least three independent experiments, and the measurements were carried out in the presence of 2 μM compounds. There are no identical letters on the superscript of the data, showing that they have statistically significant difference (*p* ≤ 0.05) between the two groups. Any letter being the same on the superscript of the data shows that they have no difference (*p* > 0.05) between the two groups. Statistical analysis was performed with ANOVA followed by Duncan’s multiple range test (DMRT).

**Table 2 molecules-27-02142-t002:** Inhibition constants (K*_i_*) of the target compounds against human AChE.

Compounds	R^1 [a]^	R^2 [a]^	*n*	K*_i_* (nM)
**7b**	Me	H	5	189.82
**7e**	OMe	H	5	65.13
**7f**	H	OMe	5	118.34
**7g**	OMe	OMe	5	152.61
**8b**	Me	H	6	49.20
**8e**	OMe	H	6	50.81
**THA**	–	–	–	31.13
**GAL**	–	–	–	61.93

Note: ^[a]^ The general structure of Coumarin–BMT hybrids is included in Figure 2.

**Table 3 molecules-27-02142-t003:** Binding energy of AChE with the compounds (kcal/mol).

Comp.	ΔE_ele_	ΔE_VDW_	ΔE_MM_	ΔG_sol_	ΔG_bind_	K*_i_* (nM)
7b	19.94	−67.54	−47.58	10.10	−37.48	189.82
8b	−4.44	−63.61	−68.05	27.62	−40.43	49.20

## Data Availability

The data presented in this study are available in the article.

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
