# Peer review of "Design, Synthesis and Bioactivity Evaluation of Coumarin–BMT Hybrids as New Acetylcholinesterase Inhibitors"

_molecules, 2022, doi:10.3390/molecules27072142_

Round 1
Reviewer 1 Report
Lines 48 and 49: delete the sentence: “AChE inhibitors can be used in the treatment of AD”. It is repetitive with the phrase from line 50.
Line 56: delete "et al9,10"
Line 68: change “(Fig. 2a)” and “(Fig. 2b)” to “(Fig 1a)” and “(Fig 1b)”
Line 69: change “(Fig. 2c)” to “(Fig. 1c)”
Line 70: change “(Fig. 2d)” to “(Fig. 1d)”
Sections 4.3 and 4.3.2 have the same name.
Line 139: Authors must describe the characteristics of Figure 3. For example, what is the meaning of the letters placed on each bar? what statistical test was used?
In Table 1 and Figure 3, the authors do not mention the statistical test used for the comparison between groups.
In Table 1 and Figure 3, I find the system used by the authors to indicate statistically significant differences confusing. They should be clearer.
On lines 118 and 158 it says: "Note: [a] The general structure of Coumarin-BMT hybrids is include in Fig. 3." However, Figure 3 does not contain molecular structures. Figure 2 contains the general structure of coumarin-BMT hybrids.
In the c chart of Figure 4, the units of the substrate concentration (S) are in µM, is this correct?
Author Response
(1)Lines 48 and 49: delete the sentence: “AChE inhibitors can be used in the treatment of AD”. It is repetitive with the phrase from line 50.
Responses: Thanks for the reviewer’s constructive suggestions. We have deleted this sentence.
- Line 56: delete "et al9,10"
Responses: Thanks for the reviewer’s constructive suggestions. "et al" in line 53 has been deleted.
(3)Line 68: change “(Fig. 2a)” and “(Fig. 2b)” to “(Fig 1a)” and “(Fig 1b)”
Responses: Thanks for the reviewer’s constructive suggestions. “(Fig. 2a)” and “(Fig. 2b)” has been changed to “(Fig 1a)” and “(Fig 1b)” in line 66 of the revised manuscript.
(4)Line 69: change “(Fig. 2c)” to “(Fig. 1c)”
Responses: Thanks for the reviewer’s constructive suggestions. “(Fig. 2c)” has been changed to “(Fig 1c)” in line 67 of the revised manuscript.
(5)Line 70: change “(Fig. 2d)” to “(Fig. 1d)”
Responses: Thanks for the reviewer’s constructive suggestions. “(Fig. 2d)” has been changed to “(Fig 1d)” in line 68 of the revised manuscript.
(6)Sections 4.3 and 4.3.3 have the same name.
Responses: Thanks for the reviewer’s constructive suggestions. The “4.3.3 The synthesis of coumarin derivatives” has been replaced by “4.3.3. General procedures for the preparation of coumarin derivatives 4a~4g” in line 256 of revised manuscript.
(7)Line 139: Authors must describe the characteristics of Figure 3. For example, what is the meaning of the letters placed on each bar? what statistical test was used?
(8)In Table 1 and Figure 3, the authors do not mention the statistical test used for the comparison between groups.
(9)In Table 1 and Figure 3, I find the system used by the authors to indicate statistically significant differences confusing. They should be clearer.
Responses: Thanks for the reviewer’s constructive suggestions. We have added detailed description about statistical test of Table 1 and Figure 3 in line 121-125 and 145-151 of revised manuscript.Statistical analysis was performed with ANOVA followed by Duncan’s multiple range test (DMRT). Any letter is the same on the superscript of the data show that they have no difference(P>0.05) between the two groups. In turn, there are no same letter on the superscript of the data show that they have statistically significant difference(P≦0.05) between the two groups.
- On lines 118 and 158 it says: "Note: [a] The general structure of Coumarin-BMT hybrids is include in Fig. 3." However, Figure 3 does not contain molecular structures. Figure 2 contains the general structure of coumarin-BMT hybrids.
Responses: Thanks for the reviewer’s constructive suggestions. We have revised accordingly, in line 118 and 171.
- In the c chart of Figure 4, the units of the substrate concentration (S) are in µM, is this correct?
Responses: Thanks. We apologize sincerely for our carelessness. We have revised accordingly, in Figure 4.
Reviewer 2 Report
The manuscript was suitable for the issue of the journal, however, some significant changed must be done.
- The author should brush up the manuscript before submission such as in the abstract, the author name, the email of authors, keywords, and contribution. Please the authors check these points for the quality check.
- For an in silico part, the author must clarify if the result comes from the molecular docking or molecular dynamics simulation as in the manuscript, this part was poorly written and unclear for the discussion. This part must be edited and added more details about the discussion, not just only result presentation. This makes this part meaningless.
- Besides, if it was a molecular dynamics simulation, the 10ns simulation would not be suffice the modeling. As a reviewer, I would suggest the 50 ns simulation should be performed.
- According to 3), the RMSD must be illustrated as either the results or supporting information to show that MD simulation well represented the dynamic behavior of drug-protein complex.
- Also it would be clear if the authors informed why only MM/PBSA was shown not both PBSA and GBSA.
- The reviewer had detected self-plagiarism and some plagiarism in this manuscript. The author must have the hard work to rewrite the manuscript prior to the next revision.
Author Response
- The author should brush up the manuscript before submission such as in the abstract, the author name, the email of authors, keywords, and contribution. Please the authors check these points for the quality check.
Responses: Thanks. We apologize sincerely for our carelessness. We have added these information in line 15, 28-29, 635-648 of the revised manuscript.
- For an in silico part, the author must clarify if the result comes from the molecular docking or molecular dynamics simulation as in the manuscript, this part was poorly written and unclear for the discussion. This part must be edited and added more details about the discussion, not just only result presentation. This makes this part meaningless.
Responses: Thanks for the reviewer’s constructive suggestions. We have added the discussion on the molecular docking in line 176-192 of the revised manuscript.
- Besides, if it was a molecular dynamics simulation, the 10ns simulation would not be suffice the modeling. As a reviewer, I would suggest the 50 ns simulation should be performed.
Responses: Thanks for the reviewer’s constructive suggestions. We have re-simulated, and the simulation results differ from that of the previous simulation, which have been shown in line 176-192 of the revised manuscript.
- According to 3), the RMSD must be illustrated as either the results or supporting information to show that MD simulation well represented the dynamic behavior of drug-protein complex.
Responses: Thanks for the reviewer’s constructive suggestions. The RMSD value of compound 7b and 8b were computed, respectively. As we can see from figure 1, the RMSD values of 7b tend to be convergent after 10ns of simulation, while the RMSD values of 8b tend to be convergent after 30ns of simulation, showing the whole systems in general were equilibrated. This part have been shown in line 195-205 of the revised manuscript.
- Also it would be clear if the authors informed why only MM/PBSA was shown not both PBSA and GBSA.
Responses: Thanks for the reviewer’s constructive suggestions. Through the literature, we know that MM/GBSA performs well for both binding pose predictions and binding free energy estimations and is efficient to re-score the top-hit poses produced by other less accurate scoring functions. We have re-simulated by MM-GBSA the simulation results differ from that of the previous simulation, which have been shown in line 176-192 of the revised manuscript. We also re-simulated by MM-PBSA, the result can see from Table 1. The two results show that there was no difference in trend.
Table 1. Calculated Binding Free Energies (kcal/mol) of 7b and 8b with AChE.
|
compd |
ΔEele |
ΔEVDW |
ΔEMM |
ΔGsol |
|
ΔGbinda |
|
7b |
14.85 |
-65.32 |
-40.08 |
38.79 |
|
-11.58 |
|
8b |
-4.44 |
-63.61 |
-68.05 |
41.60 |
|
-26.45 |
- The reviewer had detected self-plagiarism and some plagiarism in this manuscript. The author must have the hard work to rewrite the manuscript prior to the next revision.
Responses: Thanks for the reviewer’s constructive suggestions. We have made some modification on this part in line 33-56 of the revised manuscript.
Round 2
Reviewer 2 Report
The authors have addressed the point I suggest as they can.